# The Role of TRL7/8 Agonists in Cancer Therapy, with Special Emphasis on Hematologic Malignancies

**DOI:** 10.3390/vaccines11020277

**Published:** 2023-01-28

**Authors:** Maria Leśniak, Justyna Lipniarska, Patrycja Majka, Weronika Kopyt, Monika Lejman, Joanna Zawitkowska

**Affiliations:** 1Student Scientific Society of Department of Pediatric Hematology, Oncology and Transplantology, Medical University of Lublin, 20-093 Lublin, Poland; 2Independent Laboratory of Genetic Diagnostics, Medical University of Lublin, 20-093 Lublin, Poland; 3Department of Pediatric Hematology, Oncology and Transplantology, Medical University of Lublin, 20-093 Lublin, Poland

**Keywords:** toll-like receptors, TLR7/8 agonist, hematologic malignancies, immunotherapy, cancer, vaccine adjuvant, leukemia, imiquimod, resiquimod

## Abstract

Toll-like receptors (TLR) belong to the pattern recognition receptors (PRR). TLR7 and the closely correlated TLR8 affiliate with toll-like receptors family, are located in endosomes. They recognize single-stranded ribonucleic acid (RNA) molecules and synthetic deoxyribonucleic acid (DNA)/RNA analogs—oligoribonucleotides. TLRs are primarily expressed in hematopoietic cells. There is compiling evidence implying that TLRs also direct the formation of blood cellular components and make a contribution to the pathogenesis of certain hematopoietic malignancies. The latest research shows a positive effect of therapy with TRL agonists on the course of hemato-oncological diseases. Ligands impact activation of antigen-presenting cells which results in production of cytokines, transfer of mentioned cells to the lymphoid tissue and co-stimulatory surface molecules expression required for T-cell activation. Toll-like receptor agonists have already been used in oncology especially in the treatment of dermatological neoplastic lesions. The usage of these substances in the treatment of solid tumors is being investigated. The present review discusses the direct and indirect influence that TLR7/8 agonists, such as imiquimod, imidazoquinolines and resiquimod have on neoplastic cells and their promising role as adjuvants in anticancer vaccines.

## 1. Introduction

The innate immune response is a universal defense mechanism against pathogens. It is divided into two levels: -Physical (e.g., the skin and mucosa membrane) and chemical (pH and enzymes) barriers [1,2,3];-Innate immune cells (monocytes, macrophages, neutrophils, eosinophils, basophils, dendritic cells, natural killer (NK) cells, and mast cells) [3,4,5].

Innate immunity is the main contributing factor not only to acute inflammation caused by infection or tissue injury but also for the adaptive immunity activation. Innate immunity generates an inflammatory response within minutes of pathogen exposition, through muti-tasking receptors. Adaptive immunity, in contrast, requires 3 to 5 days to expand clonal lymphocytes through specific and unique receptors [5]. 

### 1.1. Pattern Recognition Receptors (PRR)

Pattern recognition receptors (PRR) play a major role in the innate immune response. These PRRs are a large group of receptors that can be located on the surface of or inside cells—most often in the membrane of lysosomes, mitochondria, and endoplasmic reticulum or in the cytoplasm [6]. Most PRRs in the innate immune system can be classified into five types based on protein domain homology: Toll-like receptors (TLRs), nucleotide oligomerization domain (NOD)-like receptors (NLRs), C-type lectin receptors (CLRs), retinoic acid-inducible gene-I (RIG-I)-like receptors (RLRs), and absent in melanoma-2 (AIM2)-like receptors (ALRs) [7]. They are localized not only on immune cells, including effector immune cells (macrophages, dendritic cells, granulocytes, and lymphocytes), progenitor cells (HSPCs), and the hematopoietic stem but also on non-specialized, non-immune epithelial and endothelial cells or fibroblasts. They initiate an intracellular signaling cascade necessary to keep tissue homeostasis and remove potentially dangerous pathogens. However, these TLR functions can be damaging and even lead to death [8].

### 1.2. Toll-like Receptors (TLR) Family

Toll-like receptors belong to the pattern recognition receptors, and they play a prominent role in the development and maintenance of the immune system. There are at least 10 members of the human Toll-like receptor (TLR) family [9]. The extracellular TLR groups (TLR1, TLR2, TLR4, TLR5, TLR6, and TLR10) are expressed on the plasma membrane, whereas the intracellular TLR groups (TLR3, TLR7, TLR8, and TLR9) are expressed in the endosome and endoplasmic reticulum. TLR4 is found on the plasma membrane as well as in the intracellular compartments [10]. The differential expression of TLRs forms the basis for many key immune responses. For example, monocytes, NK cells, mast cells, neutrophils, eosinophils, B-cells, and T-cells all have characteristic TLR expression patterns with a specific immunological effect. The expression of TLRs is modulated rapidly in response to pathogens, cytokines, and environmental stresses [11]. Generally, the TLRs expressed on the cell surface recognize microbial membrane proteins, lipids, and lipoproteins, whereas endosomal TLRs recognize microbial- and self-derived nucleic acids [12]. 

### 1.3. Structure of TLRs

Toll-like receptors belong to the type I transmembrane family, which is characterized by the extracellular leucine-rich repeats (LRR) for the recognition of pathogen-associated molecular patterns (PAMPs)/damage-associated molecular patterns (DAMPs), transmembrane domains, and intracellular toll-interleukin 1 receptor (TIR) domains for the activation of downstream signal transduction pathways [13,14]. Each LRR consists of a β strand and an α helix connected by loops. The LRR domains consist of 19–25 tandem LRR motifs, 24–29 amino acids in length, which contain the motif XLXXLXLXX and also other conserved amino acid residues (XØXXØXXXXFXXLX; Ø = hydrophobic residue) [15].

### 1.4. Ligands for TLRs

TLRs recognize related PAMPs and DAMPs. PAMPs are conserved molecular motifs. Most of them are fundamentally functional components of microbes such as LPS, flagellin, and nucleic acid (dsRNA, ssRNA, and DNA) essential for pathogen survival [16,17]. DAMPs are intracellular molecules that are released by dead cells and cells exposed to life-threatening stress. These are, among others: heat-shock proteins (HSPs), adenosine triphosphate (ATP), high-mobility group box 1 protein (HMGB1), uric acid, fibronectin, and hyaluronan fragments [18,19].

All TLRs’ subfamilies recognize different types of PAMPs. TLR7 and TLR8 recognize a uridine- and guanosine-rich single-stranded RNA (ssRNA) inside the cytoplasm and are mainly localized endosomally. This means their ligands are required to be internalized to the endosome before signaling pathways can become activated [20,21,22].

As most preclinical studies are carried out using rodent cancerous cell lines, it is important to mention that most human TLRs are closely parallel to mice TLRs. Although TLR8 has been perceived as nonfunctional in mice in the past [20,23], recent studies suggested that it does recognize a combination of imidazoquinoline and poly-T oligodeoxynucleotides [24]. The natural ligand of mouse TLR8 still remains elusive. Moreover, another novel study revealed that TLR8 is a negative regulator of neurite outgrowth and a stimulator of neuronal apoptosis in mice [25].

### 1.5. The TLR7/8 Signaling Pathway

TLR7/8 transmit signals through a myeloid differentiation primary response 88 (MyD88)-depended pathway (Figure 1). IL-1R-associated kinases 1/4 (IRAK1/4) recruit TNF receptor-associated factor 6 (TRAF6). TRAF6 activates transforming growth factor beta-activated kinase 1 (TAK1), which activates two pathways, one of which leads to the activation of the IκB kinase (IKK) complex-nuclear factor kappa-light-chain-enhancer of activated B cells (NF-κB) pathway and the other to the activation of the mitogen-activated protein kinases (MAPKs). TRAF6 can also recruit IFN regulatory factors (IRFs) and stimulate interferon signaling.

The IKK complex consists of catalytic subunits IKKα, IKKβ, and IKKγ, and it phosphorylates the NF-κB inhibitory protein IκBα, which is then degraded by the proteasome. This allows NF-κB to enter the nucleus.

TAK1 activation releases the activation of MAPK family members, then transcription factors such as cyclic AMP-responsive element-binding protein (CREB), and activator protein 1 (AP-1).

Finally, NF-κB, CREB, and AP-1 stimulate proinflammatory cytokines (e.g., IL-1, IL-6, TNFα, etc.) and chemokines (e.g., MCP-1, CCL, and CXCL chemokines) and induce the upregulation of type I and II interferons [21,22,26,27,28]. 

The human TLR7 and TLR8 genes are located on the short arm of the X chromosome (Xp22) [29]. Sex-related differences in immune response are mainly attributed to a distinct hormonal balance as well as a dissimilar genetic makeup [30]. Studies suggest that the presence of two X chromosomes plays a major role in enhancing adaptive immune responses. Such studies demonstrated that female peripheral blood lymphocytes produce considerably higher levels of IFN- α in response to TLR7 stimulation compared with male lymphocytes [31,32,33]. 

The precise mechanisms causing these sex differences in immunity are unknown. Little is known about sex-mediated variations between females and males oncological and hemato-oncological diseases. Cells’ heterogeneity may play a part in the dissimilarities in immune-mediated diseases such as systemic lupus erythematosus (SLE) and infectious diseases such as HIV infection [31,34,35]. 

Taking all of these data into consideration, it does seem plausible that increased production of the interferons in the female immune response may have a positive impact on the antitumor effect compared to the male response [36].

### 1.6. Tumor Microenvironment

The tumor microenvironment (TME) consists of many types of cells (immune cells, endothelial cells, inflammatory cells, lymphocytes, fibroblasts, and cancer cells), the extracellular matrix (ECM), blood vessels, and chemokines [37]. Innate immune cells can be both pro- and anti-tumorigenic, depending on complex cross-talk and the different chemokines in TME. The adaptive immune system, meanwhile, can specifically attack tumor cells and is regarded as being the most effective for tumor elimination [38]. Cancer-associated fibroblasts (CAFs) build the structure of the microenvironment, by synthesizing much of the ECM, and have a major impact on tumor progression and therapy [39]. The TME has a significant impact on drug penetration and function and is associated with drug resistance and low response rates [40].

### 1.7. Dual Character of TLR Signaling

Innate immune cells can, on the one hand, promote tumor growth and malignant transformation and, on the other hand, prevent tumor progression. There is a complex interaction between PRRs, immune cells, and tumor cells in the tumor microenvironment. The effects of TLRs’ stimulation depend on both the receptors and tumor types. All these features imply that TLRs can be positive and negative regulators of cancer. 

TLRs can promote tumorigenesis through various mechanisms. TLRs’ signaling pathways lead to the release of proinflammatory mediators in a tumor microenvironment that is characterized by chronic inflammation [41,42]. TLRs can also recruit and polarize more immune cells toward tumor-supporting cells to enhance immunity in the tumor microenvironment. Released cytokines impair the function of antigen-presenting cells and specific immunity, thus causing tumor immunotolerance and facilitating metastasis [43]. TLRs are activators of the NF-κB pathway, which in turn regulates the transcription of antiapoptotic genes such as *iNOS, Bcl-2, c-FLIP, IAP*, and TRAF molecules. There is increasing evidence that TLRs may provide signals to promote apoptosis evasion [44]. There is also a theory that the development of cancer may be due to an impaired tissue repair response that depends on MyD88 signaling and, thus, TLRs’ activation [45]. 

On the other hand, a positive effect of TLR activation on the tumor microenvironment was noted. For example, the usage of TLR agonists leads to the reprogramming of M2 macrophages to M1 macrophages [46,47]. Moreover, nanoparticles conjugated with specific ligands, which are inhibited by hypoxic environment and lactic acid, could target DCs and modulate their maturation and activation [48]. 

There are many other studies suggesting a protective role of TLR stimulation and therapeutic options based on TLR ligation. Here, the current literature regarding the usage of TLR7 and TLR8 agonists is reviewed.

## 2. Toll-like Receptors and Hematology

### 2.1. Role of TLRs in Hematopoiesis

The formation of blood components is a self-regenerative process that occurs all throughout human life, starting during fetal development [49]. All blood cell types originate from hematopoietic stem cells (HSCs). Hematopoietic stem cells mostly occupy the bone marrow, although some of them circulate in the blood stream or tissues. The most important positive marker for human hematopoietic and progenitor stem cells is the CD34 antigen. TLRs not only influence the adaptive immune system but also the circulation, differentiation, and action of HSCs [50]. 

Studies showed that human bone marrow hematopoietic progenitor cells constitutively express several toll-like receptors (in particular TLR4, TLR7, and TLR8) [51] and respond to receptor involvement [51,52,53,54]. Separate experiments showed that TLR9 was also displayed by CD34+ progenitor cells [51]. Other analyses revealed that many TLRs such as TLR9 and TLR10 were mostly linked to CD19+ B-cells, whereas others—TLR2, TLR4, TLR8—were expressed to the utmost in CD14+ mononuclear cells [55]. TLR7 and TLR8 can primarily be found in the myeloid cell population, including neutrophils, monocytes, and DCs with a scarce expression of hematopoietic stem and progenitor cells (HSPCs) [56]. Human pDC mainly expresses TLR7 and TLR9, whereas mDC expresses most TLR except TLR7, TLR8, and TLR9 [54]. Moreover, the common DC progenitor (CDP) expresses elevated levels of TLR2 and TLR9 and moderate levels of TLR4 and TLR7 in comparison to other HSPCs [53]. Once PAMPs from the restricted origin of an infection have spread systemically, they can span the bone marrow and be sensed by hematopoietic progenitor cells’ TLRs. TLR signaling is able to have an influence over HSPCs both by cell autonomous and cell non-autonomous mechanisms [57]. Firsthand stimulation of TLR2, TLR4, and TLR7/8 causes multipotent HSPCs, common myeloid progenitors (CMPs), and granulocyte–macrophage progenitors (GMPs) to convert to monocytes, macrophages, and DCs at the cost of other progenitors [51,53,56]. CD34+ cells incubated with TLR7/8-specific agonists such as small interfering RNAs (siRNA-27), R848 (resiquimod), and loxoribine can differentiate into myeloid cells (early as well as late granulo-monocytic progenitor cells) [51,53,54]. A significant cell population of the CD34+ progenitor increased the expression of CD11c (deliberated as one of the markers of myeloid DC in humans) and CD13 (the early myeloid marker) [51]. Other studies showed that treatment with R848 induces considerable heightening in bone marrow (BM) resident cDC, a lowering in the common dendritic cells’ progenitors and pre-DCs, and upregulation of CD83 (DC activation marker); treatment prompted an expansion of phenotypic HSC with a decreased repopulating potential and HSPC mobilization [56]. Another study found that although TLR-induced lineage skewing of upstream progenitors can give rise to multiple cells, in the case of BM DC-restricted progenitors (CDPs) TLR induction results in their mobilization via down-regulation of chemokine receptor CXCR4 expression and recruitment to lymph nodes through up-regulation of chemokine receptor CCR7 expression [53]. TLR7 and TLR8 signaling in DC is not required for the increase in DCs or HSPCs or HSPC mobilization [56].

### 2.2. TLR Agonists in Hematology

#### 2.2.1. TLR Agonists as a Treatment for Lymphomas

Lymphoma treatment options include skin-directed therapies such as topical steroids, chemotherapy, retinoids, phototherapy, and radiotherapy. For the advanced stage, systemic treatments by biological or targeted therapies are used [58,59]. A promising, new approach is represented by TLR agonists that can stimulate the immune system to induce an anti-cancer response.

In a phase 1 randomized controlled trial, the efficacy of a TLR7/8 agonist was evaluated. Twelve patients with stage IA-IIA cutaneous T-cell lymphomas (CTCL) took part in this clinical trial. They applied 0.03% or 0.06% topical resiquimod gel to a limited number of skin lesions over a period of several weeks. Clinical improvement was observed in 92% of patients, with all treated lesions clearing in 33% of patients. A reduction in the percentage of malignant T-cell clones was observed in 90% of the biopsy lesions that were in post-treatment. The successful response to treatment was associated with the increased production of IFN-γ and TNF-α by CD4+ T cells and the enhancement of NK cells’ function. Increased CD80 expression was also observed; however, the DC maturation in peripheral blood did not correlate with an increased clinical response. Three patients had a response in their untreated lesions, suggesting that systemic antitumor immunity may develop after topical therapy with these drugs [60].

Previous in vitro observations demonstrated that TLR7/8 agonists stimulate the peripheral blood mononuclear cells (PBMC) of patients with CTCL leukemia and Sézary syndrome, producing high levels of cytokines. The TLR7 agonist increased the synthesis of IFN-α, while the TLR8 agonist increased the synthesis of IL-12 and IFN-γ. The main function of induced cytokines is to enhance the cytolytic functions of NK cells and T cells by upregulating CD68 and CD25 expression. In addition, they stimulate the Th1-mediated immune response and reinforcement of cellular immunity, key elements of effective anti-cancer response. [61]. Synergistically enhancing the immune system is possible by combining IFN-γ or IL-15 with a TLR7/8 agonist [62].

The published data on the use of imiquimod in the treatment of CTLC are case reports or series involving a small number of patients (Table 1). Lesions resolved completely without recurrence within several years of follow-up [63,64]. The severe adverse effects of topical therapy with TLR7/8 agonists were usually limited to the skin. Patients reported redness, local irritation, and, in some cases, flu-like symptoms such as muscle aches, headaches, and an increased temperature. Most side effects resolved spontaneously or after a short break in treatment [60,65,66].

These results are particularly promising regarding the potential future use of TLR7/8 agonists in the treatment of lymphomas. Resiquimod and imiquimod can be effective alternatives in the treatment of skin lesions. They are safe, well-tolerated, and highly effective drugs.

#### 2.2.2. TLR Agonists as a Treatment for Leukemia

There are some in vivo and in vitro studies supporting the efficacy of TLR agonists for enhancing and directing the immunological response against specific antigens in leukemia.

Specific TLR agonists may be growth-inhibiting and pro-apoptotic in some myeloid leukemia cell lines. Imiquimod can lead to the detention of the cell cycle and the activation of apoptosis of cancer cells [68]. A similar effect was demonstrated by resiquimod [69]. TLR7/8 agonists promote the induction of AML cells by increasing the expression of mature myeloid markers, which leads to an enhanced immune response [68,69]. 

Different cells of the immune system are involved in the anti-cancer response. An increasing role in the therapy of hematological malignancies is attributed to DC cells and their subsets. They act as one of the most effective antigen-presenting cells and activate T lymphocytes, participating in the regulation of innate and adaptive immune response [70,71,72]. The use of a TLR7/8 agonist leads to higher CD40 expression and an increase in INF-β production by pDC. Treatment with INF-β led to the upregulation of CD38 expression and a greater cytotoxicity of AML cells in the presence of daratumumab (anti-CD38 antibody) [73]. Another study about developing a dendritic cell vaccine to be used for therapy of minimal residual disease in acute myeloid leukemia assessed the uptake of apoptotic leukemic cells by monocyte-derived DC (MoDC) and DC after stimulating the TLR7/8 receptor. Resiquimod increased tumor cell uptake by DC in vitro; however, in combination with cocktail of cytokines, it interfered with the maturation process and the ability of MoDC to migrate and stimulate T lymphocytes, therefore did not find therapeutic use [74].

The TLR7/8 agonist, TNF-α, and lipopolysaccharide can induce an increase in cytotoxic T cells (CTL) production and stimulate activation of DC in an in vitro culture. Single reagents induced a significantly higher expression of CD80, CD86, CD83, CD40, CD54, and HLA-DR in AML cells. The best results were achieved using a combination of all three reagents: CD80 expression on AML cells was the highest. Stimulated and activated CTL released significantly higher levels of IFN-γ and showed a greater cytotoxicity compared to the control sample. These results indicate that the combination of a TLR7/8 agonist, TNF-α, and lipopolysaccharide induces a significantly enhanced effect of antigen presentation by AML-DC [75].

NK cells are a new form of cell-mediated immunotherapy for patients with AML. Their differentiation and cytotoxicity are induced by the cytokines (IFN-γ, IL-12, IL-15, and IL-18), with production that is differently regulated by TLR agonists [76,77]. Resiquimod stimulates the significant production of IFN-γ by NK cells, of CD8+ T cells, and of IL-12 by monocytes [78,79]. Incubation of AML cells and PBMC with a TLR7/8 agonist can induce IFN-γ production, resulting in strong NK cells’ activation and increased cytotoxicity to AML cells [80]. Other studies also confirmed significantly increased levels of IFN-γ, TNF-α, and GM-CSF in R848-stimulated NK cells’ supernatants and an enhancement in their cytotoxicity [81].

With a TLR7 agonist, it is possible to enhance the efficacy of chemotherapy or facilitate the killing of cytotoxic T cells through increased tumor immunogenicity [82]. The effects of TLR7 stimulation in CLL cells vary between studies. Most of them confirm the expression of co-stimulatory molecules, the production of pro-inflammatory cytokines, and an increased sensitivity to killing by cytotoxic effectors [83,84]. Another study showed a synergistic effect between TLR7 and cladribine. The percentage of necrotic cells after imiquimod application was higher by 3.8% compared to the medium alone, and the effect was enhanced by the addition of a chemotherapeutic agent. The TLR7 agonist led to a significant increase in IL-6 and IL-10 and a decrease in IL-17A. However, there was no increase in the CD80, CD86, CD40, or CD95 on CLL cells [85].

Other studies suggested that the stimulation of TLR7/8 receptors in CLL leads to improved viability of leukemic cells. Specific TLR-7 ligands induce a marked increase in NO production in B-CLL cells, which has an anti-apoptotic effect [86].

Preclinical studies showed that TLR agonists can effectively treat AML and inhibit cancer cell proliferation. However, their potential in the treatment of CLL is questionable and requires further research.

### 2.3. Negative Aspects of TLR Signaling and Its Role in Hematopoietic Diseases

HSCs of mice treated repetitively with small doses of LPS expressed features of damage. Cells were not able to withhold quiescence; after transplantation, they were found to be myeloid skewed, and, additionally, they were not sustained in serial transplants and developed lymphoid progenitors with unsatisfying levels of E47 transcription factor [87]. Persistent TLR signaling may have subsequent consequences such as long-term harm of the self-renewal potential and functional solidity of HSC [87,88].

In patients with myelodysplastic syndromes, especially lower-risk MDS, TLR signaling is magnified. Such a phenomenon is due to a combination of genetic and epigenetic changes that influence various components of the TLR signaling pathway and have both direct and indirect effects on precancerous and cancerous cells, for instance, marked cell-death and ineffective hematopoiesis [50]. Hyper-activated innate immune signaling, enhanced TLR signaling in CD34+ HSPCs, and loss of TLR pathway repressors have been observed [89,90,91,92,93,94]. Various genes, known to be regulated by TLRs, are overexpressed or aberrantly activated in patients’ BM CD34+ cells in contrast to healthy individuals [89,90,91,92,93]. Patients’ hematopoietic and progenitor stem cells displayed sustained myeloid expansion upon continuous inflammation [93]. MyD88 expression levels have a tendency to be heightened in patients with lower-risk MDS [91]. Moreover, the activating mutation of MyD88 correlates with lymphoid malignancies [57,91].

In addition to their link to the immune cells, TLRs are also found in cancer cells, where TLR7/8 signaling is not very easy to understand and is related to either beneficial or harmful outcomes.

#### 2.3.1. B-Lymphoid Malignancies

TLR7 expression is associated mostly with B cells, DCs, and monocytes, whereas TLR8 expression is associated mostly with DCs, monocytes, and granulocytes [95]. B cells express a number of toll-like receptors (predominantly TLR1, TLR6, TLR7, TLR9, TLR10), and a TLR-initiated reaction in such cells leads to expansion, an anti-apoptosis effect, and PC differentiation [96]. Other studies show that TLR7 and TLR9 play a vital role in the autoreactive B-cells’ activation [97].

B-cell homeostasis is largely regulated by B cell activating factor (BAFF) and a proliferation-inducing ligand (APRIL) ligands and their receptors (B-cell maturation antigen/BCMA, transmembrane activator and calcium-modulator and cytophilin ligand interactor/TACI, and B cell-activating factor receptor/ BAFF-R). The BAFF–APRIL system is greatly engaged with the selection of conditions such as multiple myeloma/ MM (since BCMA promotes myeloma growth in BM) and autoimmune diseases. Experiments showed that pDCs in human blood carry the BCMA protein and display it on the cell surface upon TLR engagement (TLR7/8 agonist R848 and TLR9 agonist CpG-A). Agonists induced the release of serum B-cell maturation antigen (sBCMA) from pDCs and IFN-α (in comparison with the TLR9 agonist, R848 did not influence the levels of interferon). Drugs targeting such receptors are under development and can have positive outcomes on the treatment of lymphoid neoplasms [98].

TLRs were proclaimed to have been found on freshly isolated myeloma cells and the MM cell lines, with their expression being notably higher than in normal plasma cells [96,99,100,101]. TLR-specific ligands induce intensified proliferation, endurance, cytokine and chemokine excretion, induction of apoptosis or protection from it, drug resistance, and the immune escape of the MM cell lines, in some measure due to autocrine interleukin-6 production [96,102]. Human myeloma cell lines (HMCL) express a broad range of TLRs at the gene and protein levels, with TLR1, TLR4, TLR7, TLR8, and TLR9 being the most detectable [99,100]. Novel studies found TLR7 to be the most frequently expressed of all the TLRs in the MM cell lines, with some of its ligands such as loxoribine and R848 inhibiting apoptosis and promoting the proliferation of cell survival [101].

Furthermore, TLRs are associated with CLL. Leukemic cells from CLL patients express TLR1, TLR2, TLR6, TLR7, TLR9, and TLR10. The sequence of expression is similar to the one detected in memory CD19+ CD27+ B-cells, although TLR-2 and TLR-7 are expressed at higher levels in B-CLL [103,104]. After stimulation of B-CLL cells with TLR agonists, the upregulation of activation markers (CD40, CD86, CD80, and HLA-DR molecules) was observed, just like in normal B-lymphocytes. In contrast to other agonists, stimulation by loxoribine has only a negligible effect on leukemic cells’ survival (induction of apoptosis as well as proliferation of a subpopulation of B-CLL cells) and has no effect on normal B cells, while cancerous cells produce significantly higher levels of TNFα. TLR-9 and TLR-7 stimulation of CD38 B-CLL leads to an increased numbers of cells expressing CD38 (surface protein with expression that increases upon normal B-cell activation, a marker of disease aggression in B-CLL) on sorted B-CLL cells [104].

MyD88 mutations arise in a number of human malignancies such as Waldenstrom macroglobulinemia (WM), CLL, cutaneous B-cell lymphoma (CBCL), and primary central nervous system lymphoma (PCNSL) [105,106]. Improper activation of TLRs has been featured in activated B-cell-type diffuse large B cell lymphoma (ABC-DLBCL), which has a pathogenesis that centers around constitutively active NF-κB. The MyD88 L265P oncoprotein attaches to TLR7 and TLR9, increasing the signals from such receptors as a consequence. Suppression of TLR7 or TLR9 promotes apoptosis among ABC-DLBCL cell lines. A reduction in the proteins essential for TLR7 and TLR9 trafficking and signaling, as well as the pharmacological inhibition of receptors’ function, reduces the survival of cancerous lines. A better understanding of the oncogenic mechanisms of MyD88 mutations provides the reasoning for targeting TLR7 and TLR9 signaling for ABC-DLBCL therapy [105].

The connection between enhanced TLR expression and hematopoietic malignancies indicates there are therapeutic benefits from TLR signaling repression. Various novel TLR pathway inhibitors emerge, such as HJ901 (TLR7/9 inhibitor) [107] and CA-4948 (IRAK4 inhibitor) [108]. They are becoming the therapeutic targets for the treatment of MDS, AML, ABC-DLBCL, and non-Hodgkin’s lymphoma, among other diseases.

#### 2.3.2. Myeloid Malignancies

AML blasts express a vast range of TLRs [69,109,110]. Higher mRNA expression of TLR pathway molecules, the upregulation of negative TLR signaling pathway regulators and NF-κB inhibitors, and lower expression of transcriptional regulators were observed, which are thought to correlate with increased neoplastic cells’ survival [109]. A number of TLRs are also expressed in AML cell lines, and particular receptor agonists influence some cell lines and induce growth inhibitory and apoptotic effects [69,109]. R848 restrains the growth of human AML cells in immunodeficient mice, within a direct effect on cancerous cells [109]. TLR8 stimulation can have a straight anti-leukemia activity as well. Treatment with R848, mediated through TLR8, prompts terminal differentiation and inhibits AML proliferation; in addition, it harms cell and colony growth in vitro and tumor formation in vivo [69]. Different studies showed that in comparison to other TLR agonists, imiquimod profoundly delays the propagation of all tested AML cell lines, and an aggregation of cells in the M phase was observed [110].

The targeting of innate immune signaling, particularly TLRs, is an attractive new therapeutic approach, although the specific contribution of such signaling to disease progression is not clear.

## 3. TLR Agonists in Oncology

The number of oncological patients is growing every year; therefore, it is necessary to develop more and more effective treatment methods. In recent years, much interest has been aroused by immunotherapy, and, along with it, TLR agonists have been considered as a good target for anticancer therapy.

### 3.1. Induction of Systemic Immunity to Neoplasms

Systemic exertion of TLR agonists may cause perilous effects such as cytokine release syndrome (CRS) or systemic autoimmunity [111,112]. CRS is a systemic inflammatory reaction that can be induced by many factors, such as infections and medications. It represents one of the rifest considerable side effects of therapies with T-cell-engaging immunotherapeutic agents. Severe cases include significant hypotension, circulatory shock, vasculatory leakage, disseminated intravascular coagulation, and multi-organ system failure [113]. In addition, less dangerous reactions were also noticeable in patients, such as fever, fatigue, back pain, headache, shivering, and lymphopenia [114]. Therefore, the topical or intratumoral usage of these drugs is considered, especially in combination with other substances such as immune checkpoint inhibitors or cytostatics [115,116]. Despite the local use of these drugs, systemic effects have been noticed, but they seem to be useful in this case. This is about systemic immunity and possibly an anti-metastatic role. In murine models, after an intratumoral injection of TLR agonist in monotherapy or in combination with checkpoint inhibitor, tumor growth was suppressed both in the primary injected and remote noninjected sites, suggesting the induction of systemic immunity. As a result of the intratumoral TLR7 agonist treatment, the number of M1 macrophages was increased, while the number of M2 macrophages associated with the tumor was reduced. In addition, the infiltration of CD8+ T-cells was promoted, resulting in the increased production of cytokines. Adding anti-PD-L1 to this therapy resulted in the increased clonality of the T-cell receptor in tumor T cells. This type of therapy induced a specific adaptive antitumor immune response [115]. Systemic effects have also been shown in patients after melanoma treatment with a local application of TLR agonist and monobenzone. This combination of medicaments induced the partial regression of metastases and a vitiligo-like reaction in patients with melanoma. Actually, monobenzone itself can cause vitiligo, so a collateral immune adjuvant may be essential to consolidate the systemic immunological reaction to melanocytes and, thus, an anti-melanoma effect [117]. This type of treatment may be a correct option for patients with difficult-to-treat metastatic tumors and could be the next step in the development of non-invasive anti-cancer therapy.

### 3.2. TLR Agonists as a Non-Invasive Alternative to Surgery

There is a discussion of the potential benefits and risks of treatment options in patients with vulvar high grade squamous intraepithelial lesions (vHSIL), including the oncological safety of topical imiquimod treatment in comparison to the surgical methods (excision or ablation) that are currently standard. With vHSIL, we are dealing with an immunosuppressive environment; it consists of the percentage of M2 macrophages, which significantly outstrip the percentage of M1 macrophages. In addition, there is an increased number of regulatory T cells in the area of the lesion. The usage of imiquimod is associated with the normalization of the ratio of CD4+ T-cells to CD8+ T-cells as well as the stimulation of dendritic cells to secrete pro-inflammatory cytokines, resulting in immune infiltration. Such actions’ outcomes relate to the histological regression of vHSIL. Nevertheless, combining these surgical methods with imiquimod does not seem to reduce the risk of recurrence, though it may allow less extensive excisions and anatomical and functional conservation of the vulvar structures [118]. Studies demonstrated that imiquimod is a secure and efficient method for women with vHSIL, so it can be considered as a first-line treatment, but it demands a good patient compliance due to its relatively long treatment period [119].

Similar considerations apply to vulvar Paget’s disease. The present-day first-line treatment for in situ vulvar Paget’s disease is local excision, but a significant problem is that the extension of the disease is commonly wider than what is apparent in the skin [118]. Recurrence rates range from 15% to 70% and may require repeated surgical treatment, and vulvar Paget’s disease is capable of having psychosexual effects on patients and decreasing their quality of life [120]. Studies showed a clinical response in most patients with noninvasive vulvar Paget’s disease after a 5% imiquimod cream therapy; therefore, this seems to be a sensible, safe conservative treatment option in comparison to invasive surgical methods [120,121].

There are situations when clinical or patient factors preclude or impede the surgical excision of a neoplastic lesion, including melanoma. These comprise the location of the lesion, its size, and the patient’s general condition. In such situations, it seems reasonable to try to use another treatment, including topical imiquimod in monotherapy or in combination with other substances acting synergistically or enhancing drug penetration, such as 5-fluorouracil, tazarotene, or tretinoin. A number of case studies reported on the use of this type of treatment in melanoma in situ and showed that it was effective in some of the treated patients. After a cycle of imiquimod usage, biopsies were taken, and they demonstrated the absence of neoplastic lesions [122,123]. Moreover, studies demonstrated that therapy with topical imiquimod may be used as a primary treatment in patients with lentigo maligna instead of surgery [124]. This shows there is another therapy option for patients who are not able to undergo standard surgical excision.

### 3.3. TLR Agonists as a Treatment for Residual or Recurrent Disease

Sometimes, patients after the surgical excision of a neoplastic lesion have positive margins with cancer cells, but they either are not adequate candidates for reresection or desire to avoid another surgery for other reasons. For instance, residual disease in the head and neck is an immense trouble due to the difficulties with complete resection because of the location of the lesion, which may cause tremendous esthetic or functional deterioration. This demonstrates the need for a non-surgical method of treating residual disease. Some studies showed that topical imiquimod is an effective alternative to re-excision in selected people with positive MIS (melanoma in situ) margins, after an initiatory surgical resection of melanoma. Over 90% of patients had complete regression of their disease after a cycle of imiquimod treatment [125,126]. This may be related to the pro-apoptotic effect of imiquimod on melanoma cells. There are reports of imiquimod-induced apoptosis being dependent on Bcl-2 expression. The potential of using topical imiquimod to induce apoptosis in a tumor-selective manner for the treatment of melanoma and its metastases has been demonstrated. Interestingly, resiquimod, despite its similar structure to imiquimod, does not show a pro-apoptotic effect in melanoma cells [127]. Studies also showed that imiquimod therapy may be used as an adjuvant therapy after surgery in patients with lentigo maligna, which is capable of reducing recurrence rates [124,128].

Due to reports of the effectiveness and security of imiquimod treatment for vulvar lesions, research was conducted in patients with residual or recurrent CIN lesions (cervical intraepithelial neoplasia) after primary surgery. The evolution of non-invasive treatment methods and their application may protect patients from repetitive surgeries and the complications associated with them, such as hemorrhage, infection, or pregnancy problems [129]. In this situation, imiquimod treatment turned out to be less effective than LLETZ (large-loop excision of the transformation zone), which is now the treatment of choice, but it could be an alternative for patients who do not want to undergo surgical treatment [130,131].

## 4. TLR Agonists as Future Targets in Cancer Therapy

In order to prevent side effects after the systemic administration of TLR agonists, new methods of administering them are currently being researched to reduce the exposure of the patient’s healthy tissues to drugs and to better confine the drugs to the tumor. For this purpose, the use of liposomes [132,133,134], hydrogels [116,135], immune implants [136], and pH-responsive nanoparticles [137], which include a TLR agonist alone or in combination with other drugs, is being investigated.

In recent years, the combination of immunotherapy using TLR agonists with other methods has attracted a lot of interest, such as sonodynamic therapy [132,138] and photothermal therapy [135,139,140] (Table 2).

Cancer vaccines have therapeutic potential in the treatment of hematological diseases. Their use with TLR agonists shows an enhanced antitumor effect. The main mechanism to achieve this is through the stimulation of tumor CTL [141,142]. A synergistic effect was observed when combining TLR agonists with monoclonal antibodies [143,144] and inhibitors of cell-cycle-regulating pathways [145]. Of particular interest is the delivery of chemotherapeutics and immunotherapeutics. Novel combinations generate strong cytokine production and enhance the CTL response, leading to the eradication of both local and distant tumors [146]. In addition, TLR agonists can be used in the treatment of anemia by increasing the production of erythroids from the myeloerythroid precursors and progenitors in the bone marrow [147].

A new path in research is the use of TLR stimulation on chimeric antigen receptor (CAR) T-cell therapy. This has turned out to be helpful in the treatment of hematological malignancies [148], so efforts are being made to also achieve such an effect in solid tumors. However, this is hampered by the depletion of CAR T-cells. In this case, the stimulation of the signaling pathway associated with the TLR turned out to be correct, and, therefore, the CAR T-cells were reactivated, as evidenced by the reduction in exhaustion markers, PD-1+Tim-3+, as well as tumor shrinkage [149]. Despite this success, further research in this direction is needed to better understand the interaction of the individual components of the signaling pathways, in order to obtain the best models for anticancer therapy.

## 5. Conclusions

The contribution of TLR7 and TLR8 to non-specific immunity has been broadly examined, including the role of TLR signaling in both immune cells and in cells’ precursors. Moreover, TLRs were identified in neoplastic cells, in which such signaling results in either tumor inhibition or progression.

The latest studies show promising results, after the use of TLR agonists as an activator of innate immunity, in the fight against tumor growth and metastases, so their incorporation in new treatment methods should be considered. There may turn out to be a bright future in modern anticancer therapy. In many studies, TLR agonists proved to be a good tool for cancer immunotherapy, but there are still situations where standard treatment methods are more effective. Therefore, further research in this direction is needed to ensure that the new treatment regimens offered to patients are at the highest level of efficacy, both in terms of enhancing the antitumor response and safety for patients with the lowest number of adverse effects.

## Figures and Tables

**Figure 1 vaccines-11-00277-f001:**
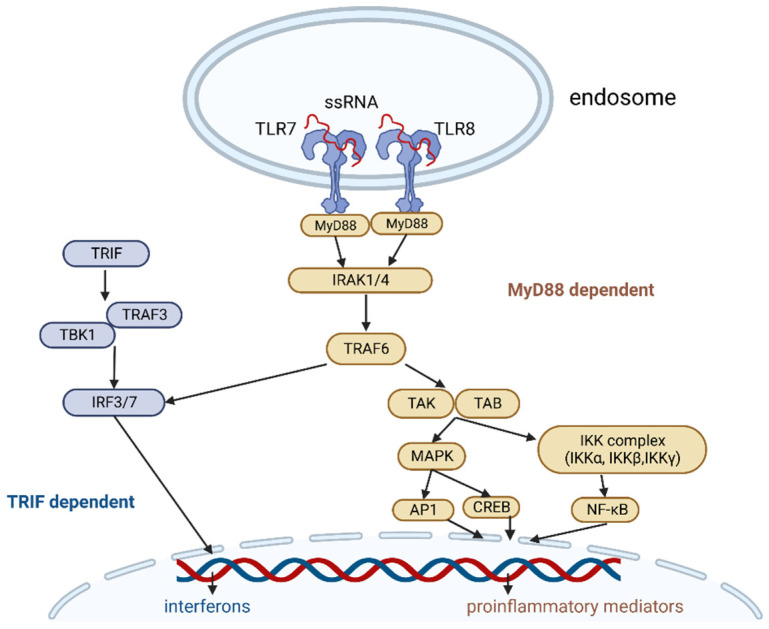
TLR7 and TLR8 signaling pathway. Image created with biorender.com (accessed on 19 December 2022). Abbreviations: ssRNA—single-stranded ribonucleic acid, Myd88—myeloid differentiation primary response 88, IRAK1/4—IL-1R-associated kinases 1/4, TRAF6—TNF receptor–associated factor 6, TAK—transforming growth factor beta-activated kinase 1, TAB—TAK1-binding protein, MAPK—mitogen activated protein kinase, IKK—IκB kinase, AP1—activator protein 1, CREB—cyclic AMP-responsive element-binding protein, NF-κB—nuclear factor kappa-light-chain-enhancer of activated B cells, TRIF—TIR-domain-containing adapter-inducing interferon-β, TRAF3—TNF receptor-associated factor 3, TBK1—TANK-binding kinase 1, IRF3/7—IFN regulatory factor 3/7.

**Table 1 vaccines-11-00277-t001:** Case series’ reported results on the use imiquimod in the treatment of cutaneous lymphoma.

Age/Sex of Patients	Disease	Intervention/ Treatment	Therapy Effect	Adverse Events	References
**66 years/** **Female**	Adult T-cell lymphoma	Imiquimod 5% cream	Nearly complete resolution of the plaque	Not reported	Messer et al. [63]
**65 years/** **Female**	Mycosis Fungoides	Imiquimod 5% cream	Nearly complete resolution of the plaques	Significant irritation, flu-like symptoms	Lewis et al. [65]
**77 years/** **Female**	Anaplastic large cell lymphoma	Imiquimod 5% cream Mupirocin 2% cream	Complete resolution of the plaque	Not reported	Kubicki et al. [64]
**68 years/** **Male**	Mycosis Fungoides	Imiquimod 5% cream Bexarotene 150mg	Complete resolution of the plaques	Not reported	Lewis et al. [65]
**21 years/** **Female**	Folliculotropic Mycosis Fungoides	Imiquimod 5% cream	Complete resolution of the plaque	Site irritation, pruritus, ulceration	Schalabi et al. [67]

**Table 2 vaccines-11-00277-t002:** Examples of studies using TLR agonists in combination with sonodynamic or photothermal therapy.

TLR Agonist	Components of the Tested Molecules	Activating Factor	Examined Cell Lines	Additional Component of Therapy	Reference
R837	Liposomes with R837 and HMME	Biomedical ultrasound	4T1 murine breast cancer cell line and CT26 murine colorectal line	anti-PD-L1	[132]
R837	Nanoparticles made of amphiphilic polymer C18PMH-PEG with R837 and MB	Biomedical ultrasound	HUVECs—human cell line, CT26 murine colorectal cancer cell line, 4T1 murine breast cancer cell line, SKOV3—human cell line	anti-PD-L1	[135]
R837	Nanocrystals with PDA and R837	NIR laser irradiation	B16-F10 murine line	-	[138]
R848	Nanoparticles made of PLGA with ICG and R848	NIR laser irradiation	PC-3, LNCaP, DU 145—human prostate cancer cell lines, RM9 murine prostate cancer cell line	-	[139]
R848	Nanoparticle based on the surgical tumor-derived cell membranes with R848 and MPDA	NIR laser irradiation	4T1 murine breast cancer cell line, CT26 murine colorectal cancer cell line	anti-PD-L1	[140]

R837—Imiquimod, R848—Resiquimod, HMME—hematoporphyrin monomethyl ether, MB—methylene blue, PLGA—poly(lactic-co-glycolic acid), ICG—indocyanine green, PDA—polydopamine, MPDA—mesoporous polydopamine, NIR—near infrared, HUVECs—normal human umbilical vein endothelial cells, SKOV3—human ovarian carcinoma cells.

## Data Availability

Not applicable.

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
