# Peer review of "The Role of TRL7/8 Agonists in Cancer Therapy, with Special Emphasis on Hematologic Malignancies"

_vaccines, 2023, doi:10.3390/vaccines11020277_

Round 1
Reviewer 1 Report
Dear Authors,
The review manuscript contains valuable and useful information and in general is very interesting to read. However, there were sentences where I had trouble figuring out what you were trying to say because the English grammar was not correct. Also, some of the words you chose to get your message across seemed incorrect. There were too many places like this in the manuscript and I did not want to change your meaning. Therefore, I suggest that the manuscript undergo English language correction before it gets further reviewed.
Reviewer 2 Report
Writing is uneven in organization and phrasing. English editing needed. Some phrasing has grammar issues, some phrasing is unclear. Often there is missing or inappropriate indefinite articles, prepositions and pronouns. The use of conjugations of the verb “to be” is sometimes incorrect. Overall, it is a bit redundant and could be written more tightly.
Section on dual role is not well written. Admittedly complex issues to address are not easy but needs to be done better. That section needs a lead sentence or 2 laying out that issue and its complexity. It should also be better connected to the dual role of innate cells that can be pro or antitumor.
Should note that TLRs are quite similar but not identical in mice where most of the TLR study has been done, and TLR8 has a different function in mice than in humans.
Line 24, abstract, what are “costimulatory particles”
Some acronyms not defined or not defined until after used previously.
RP105 is inserted with no previous background in line 166, it is in the family but not usually listed as a TLR, so using means it needs to be explained or removed
Table 1 needs better focus on type I pathway vs inflammatory pathway, this makes it seem like it is all one pathway
Line 123 does not list cancer cells with the TME
Line 183 mentions “3 fold heightening” but is not clear what that refers to and what was done to make it happen
Line 214, listed cytokines do much more than activate NK, what the NK cell shown to be important?
Line 215-220 is in vitro and not very relevant and poorly explained and integrated.
Section 2.4, the article is on TLR 7,8 agonists, why are antagonists being discussed?
Table 2, note which cell lines are mouse/human
Section on leukemia tends to list specific studies and not provide context such as whether this was in patients or in vitro. It tends to be an extensive list of specific facts that are poorly connected to whether it matters in the disease.
unclear writing Line 188
line 496, CLT not defined
Reviewer 3 Report
In this manuscript, authors aim to review the role of TLR7 signaling and TLR7 agonism, with a focus on hematology. While the paper is interesting and starts of quite strong, the manuscript needs a bit more work in the end for it to be impactful. Please find my recommendations below.
Major
1. In general, I think the first couple of sections are very explanatory, but starting at section 3, the actual information contained in the manuscript becomes less in quality and/or fewer examples are provided. Authors should strive to improve on this.
2. Section 4 – what I am missing here is the discussion of how TLR signalling can be exploited outside of agonists, i.e. through chimeric receptors (PD-L1/TLR7 or PD-1/TLR7) or chimeric antigen receptors (TCR/TLR7). Please consider adding this, even if it is just speculative. Now, only molecule agonists are discussed.
3. Please consider having a (near) native English colleague or an editing service to edit the manuscript to improve the level of English.
Minor
General
I would suggest authors to add information regarding sex-differences, as TLR7 specifically is expressed differently between men and women, and is associated with disease.
Specific
Line 16 – remove “an”
Line 17 – suggest “in endosomes” instead of “on”.
Line 19 – change the comma to a full stop and start a new sentence.
Line 39 – is not only the main
Line 45 – “Pattern recognition molecules (PRM) play a major role in the innate immune response” would work better. Furthermore, I would suggest to substitute PRM with PRR, as it’s not a common abbreviation used in this setting, and you continue on PRRs later on anyway.
Line 60 – “play a prominent role”
Line 68 – what about T cells? See e.g. https://doi.org/10.3389%2Ffbioe.2022.1027619 for data on human CD8+ T cell expression patterns.
Paragraph staring at line 73 – this is a nice piece of history but doesn’t add any value to the review. Consider removing it.
Line 83 – PAMPs/DAMPs has not been used before – it comes in the next section. Please use unabbreviated.
Line 91, 105 – this is a bit of a singular statement, now isn’t it? Perhaps authors meant to continue on with the paragraph – consider linking this sentence with the next paragraph.
Line 99 – remove are
Line 101 remove they
Line 101 – are required internalized instead of need internalization
Line 117 – TLR7 signalling also induces the upregulation of the type 2 IFN IFN-g. Please amend.
From here on out, I will stop listing minor changes, as they are becoming too numerous – please consider having a (near) native English colleague or an editing service to edit the manuscript to improve the level of English.
Line 138 – the way the sentence is written now it states that TLRs recruit more immune cells, thus enhancing immunity, and therefore also enhancing immune evasion – I don’t think this is what authors meant, please consider rephrasing.
Line 139 and onward – this is repetitive of earlier parts of the test.
Line 151 – this last sentence is a bit too unsure. It would be better to state “Here, the current literature regarding X is reviewed.”
Line 160 – rotation? Do authors mean circulation?
Section starting at line 269 – TLR7 agonists also induce IFN-g production by CD8+ T cells, see https://doi.org/10.4049/jimmunol.1801026.
Round 2
Reviewer 1 Report
Dear Authors:
Your manuscript is very much improved over the previous version. However, there are some improvements that can be made.
Major Points.
1. In lines 233-235 the sentence reads like the train of thought was lost before the end of the sentence. I cannot figure out what you were trying to say, so I cannot fix it. To help you out I looked up the paper that you were referencing, which I believe was #54. Here is my take on the message this reference conveys. Although TLR-induced lineage skewing of upstream progenitors can give rise to multiple cells, in the case of BM DC-restricted progenitors (CDPs) TLR induction results in their mobilization via down-regulation of CXCR4 expression. Mobilized CDP offspring then migrate to LNs through up-regulation of CCR7 expression. In this manner TLR-induction assures that the number of DCs in the LNs is adequate during an ongoing immune response.
2. In lines 295-298 the description of the results presented by reference 75 is rather miss leading. Reference 75 is about developing a dendritic cell vaccine to be used for therapy of minimal residual disease in acute myedoid leukemia. This should be mentioned by you. Also, indicate that the loading of the MoDCs with the apoptotic leukemia cells was performed in vitro. This is what I think the message of this paper is. The monocyte-derived dendritic cells (MoDCs) are loaded with the apoptotic leukemic cells in vitro and then matured in vitro during a second incubation in the presence of a cytokine cocktail (CC). Although R848, the TLR agonist, was tried for these experiments because it did enhance loading, it was found that no benefit was obtained by adding it since it interfered with the DC maturation process.
3. References 54 and 75 were the only two that I checked. If there were problems with the ones that I checked, there are probably others that need to be corrected as well. Please go through your whole manuscript to make sure that you are accurately describing the results of the reports that you are referencing.
Minor Points.
1. In line 25 replace the word “particles” with “surface molecule”.
2. In line 51 change the sentence to “These PRRs are a large group of receptors that…”
3. In line 62 change the sentence to “However, these TLR functions can be ….
4. In line 260 change “(IL)-12 to IL-12.
5. In line 299 add the word “in” before cytotoxic.
6. In line 312 change “Il-12” to “IL-12”.
7. In line 348 add the word “cells” after CD34+.
8. In line 369 change the word “discovered” to “showed”.
9. In line 431 define sicj.
10. In line 435 change “a lot of” to “much”.
11. In line 437 change “direction in” to “target for”.
12. In line 443 delete the word “even”.
13. In line 474 replace the word “overstep” with “outstrip”.
14. In line 499 replace the word “reports” with “studies”.
15. In line 537 replace “accumulate the drugs” to “confine the drugs to”.
16. In line 580 change “In a lot of research” to “In many studies”.
Reviewer 2 Report
Review 2, use of costimulatory “particles” in abstract or anywhere is wrong, they are molecules, not particles
Line 100, studies are not done with rodent embryos, “embryos” is wrong
Why was a new section 1.7 of sex differences in TLR response inserted? Was this requested by another reviewer. It is not within the focus and overall is a list of disconnected findings that does not resolve into any clear or useful ideas for readers. Make it useful and relevant to the topic or remove it.
This effort to correct section 2.2.1 is still not adequate:
The 260 induced cytokines have the capacity to enhance the functions pf NK cells and T cells by 261 upregulating CD68 and CD25 expression [62].
This implies that this is the only impact of those cytokines on these cells which is not at all true.
Reviewer 3 Report
I would like to thank the authors for incorporating most/all of my comments. I have no further comments except for two minor textual changes that can be corrected during proofreading. Please find these below.
Line 109 – there’s an extra space after the dash.
Line 312 – IL (capitalized L).
